# Fuzzy Adaptive Passive Control Strategy Design for Upper-Limb End-Effector Rehabilitation Robot

**DOI:** 10.3390/s23084042

**Published:** 2023-04-17

**Authors:** Yang Hu, Jingyan Meng, Guoning Li, Dazheng Zhao, Guang Feng, Guokun Zuo, Yunfeng Liu, Jiaji Zhang, Changcheng Shi

**Affiliations:** 1School of Mechanical Engineering, Zhejiang University of Technology, Hangzhou 310023, China; huyang@nimte.ac.cn (Y.H.); mengjingyan@nimte.ac.cn (J.M.); zhaodazheng@nimte.ac.cn (D.Z.); liuyf76@zjut.edu.cn (Y.L.); 2Ningbo Cixi Institute of Biomedical Engineering, Ningbo 315300, China; liguoning@nimte.ac.cn (G.L.); fengguang@nimte.ac.cn (G.F.); moonstone@nimte.ac.cn (G.Z.); zhangjiaji@nimte.ac.cn (J.Z.); 3Ningbo Institute of Materials Technology and Engineering, Chinese Academy of Sciences, Ningbo 315201, China

**Keywords:** end-effector rehabilitation robot, assist-as-needed, fuzzy logic, potential field, human–robot interaction

## Abstract

Robot-assisted rehabilitation therapy has been proven to effectively improve upper-limb motor function in stroke patients. However, most current rehabilitation robotic controllers will provide too much assistance force and focus only on the patient’s position tracking performance while ignoring the patient’s interactive force situation, resulting in the inability to accurately assess the patient’s true motor intention and difficulty stimulating the patient’s initiative, thus negatively affecting the patient’s rehabilitation outcome. Therefore, this paper proposes a fuzzy adaptive passive (FAP) control strategy based on subjects’ task performance and impulse. To ensure the safety of subjects, a passive controller based on the potential field is designed to guide and assist patients in their movements, and the stability of the controller is demonstrated in a passive formalism. Then, using the subject’s task performance and impulse as evaluation indicators, fuzzy logic rules were designed and used as an evaluation algorithm to quantitively assess the subject’s motor ability and to adaptively modify the stiffness coefficient of the potential field and thus change the magnitude of the assistance force to stimulate the subject’s initiative. Through experiments, this control strategy has been shown to not only improve the subject’s initiative during the training process and ensure their safety during training but also enhance the subject’s motor learning ability.

## 1. Introduction

With the aging of the population and changes in lifestyle and dietary habits, the incidence of stroke is increasing annually and affecting younger individuals. Stroke can lead to brain damage, which causes a significant number of people to experience motor impairment, which is the primary cause of long-term disability [1]. In particular, the motor function of the upper limb is greatly affected, rendering individuals unable to perform activities of daily living independently. Restoring the functional ability of the upper limbs in patients with disabilities is imperative, as individuals require the use of their upper limbs to carry out various daily activities [2]. As per the neuroplasticity hypothesis, consistent and intensive training can modify the structure and function of the nervous system and restore the connection between the affected limb and the injured central nerve of the brain, thereby regaining control over the limb’s motor function [3].

In recent decades, a plethora of rehabilitation techniques have been developed to facilitate the recovery of motor functions, thereby enhancing patients’ quality of life [4,5,6,7]. They are usually based on the principles of motor skill learning to promote plasticity in motor neural networks, which requires patients to perform intensive, repetitive, task-oriented motor training. Rehabilitation robots can meet the high-density repetitive movement requirements of patients with limb disabilities, complete task-oriented rehabilitation training, and effectively promote the recovery of patients with limb disabilities [8]. Numerous clinical studies have demonstrated the efficacy of upper limb rehabilitation robots in restoring motor function in the upper limb [9,10] and even improve patients’ activities of daily living (ADLs) [11].

Rehabilitation robots are typically categorized into two main types based on their connection to the user: end-effector type robots and exoskeleton type robots. In the end-effector type [12,13,14], the robot’s end-effector is connected to the user’s limb. In contrast, the exoskeleton type [15,16,17], also known as wearable rehabilitation robots, is wrapped around the human body, and each joint is controlled independently. However, the complex anatomy of the human upper limb generates additional interaction forces and moments between the exoskeletal robot and the limb during rehabilitation training [18]. In comparison to traditional rehabilitation training methods, upper limb rehabilitation robots that are end-effector-based have distinct advantages [19]. Clinical comparison trials have demonstrated that end-effector rehabilitation robots are more effective in promoting active participation among patients with mild and severe stroke [20], as opposed to exoskeletal upper limb rehabilitation robots.

One of the most well-known robotic systems in upper limb rehabilitation research is the MIT-MANUS, which was developed by Krebs et al. [21]. This robot was designed to aid in shoulder and elbow rehabilitation, with three degrees of freedom, and it offers passive training, active-assisted training, and resistance training. The system employs an impedance control method, which allows for various applications based on the target trajectory. Vahid et al. [22] presented a new design concept for hand tendon injury rehabilitation robots. The designed system is a single-actuator device with the capability of applying rehabilitation training on all the joints of the fingers except thumb by using an exotendon network which is planted on the surface of a soft glove.

However, conventional controllers have fixed parameters that cannot be adapted to changes in the environment or changed according to the patient’s status. Alireza et al. [23] proposed a novel design of a hand rehabilitation robot and a fuzzy sliding mode controller to compensate for the effects of uncertain parameters and chattering phenomenon. However, with different patients, there are different interaction forces between the hand and the robot. Therefore, there is a requirement for adapting the parameters, and they designed an adaptive controller beside the fuzzy sliding mode controller to eliminate the effects of these forces [24]. Xu et al. [25] proposed a newly developed adaptive impedance controller based on evolutionary dynamic fuzzy neural network, where the desired impedance between robot and impaired limb can be regulated in real time according to the impaired limb’s physical recovery condition. Nevertheless, these controllers are mainly optimized for tracking performance and do not take into account the patient’s rehabilitation needs.

For upper limb rehabilitation, different rehabilitation strategies are required because the severity of injury varies from patient to patient. Passive training strategies are mainly used for patients in the acute phase of stroke when the patient’s arm is completely flaccid and unable to perform movements. However, passive training strategies may not be effective in subjects who have regained partial upper limb motor function [26]. For this group of patients, robots can provide minimal intervention that can stimulate and promote the neuroplasticity of the patient’s brain. 

Therefore, for patients in the intermediate to advanced stages of rehabilitation who have regained partial upper limb motor function, it is crucial that the robotic assistance provided is limited to only when the patient is unable to complete the task on their own. Completely restricting patients from following a predetermined trajectory to complete their rehabilitation provides no benefits. Thus, a controller that enables the patient to optimize the rehabilitation effect is needed. Such a controller, referred to as an assist-as-needed (AAN) controller or assistive/corrective controller, can offer the necessary assistance or correction to complete the rehabilitation training task in accordance with the patient’s functional ability [27].

Currently, AAN controllers have been applied to upper [28,29,30] and lower [31,32,33] limb rehabilitation to stimulate active patient participation in rehabilitation training. AAN controllers for virtual channels based on task trajectories have also been developed [26,27]. These controllers do not provide assistance when the tracking error is within the virtual channel boundaries, but apply a recovery force to assist the subject in returning to the tunnel when the tracking error exceeds the virtual channel boundaries. Guang et al. [34] proposed an AAN controller based on impedance control to provide assistance forces when the patient deviates from a predetermined trajectory during rehabilitation training. Pehlivan et al. [35] developed an upper limb robot system with an AAN controller based on disturbance observer. This controller can eliminate estimated inputs from users and provide adjustable tracking errors. Pan et al. [36] proposed a new fuzzy logic-based safety monitoring method for upper limb rehabilitation robots that responds to emergencies based on the physical condition of the damaged limb and uses position and velocity tracking errors as inputs to the fuzzy logic to change the assisted force magnitude. However, these AAN controllers only focus on the patient’s position tracking performance and ignore the patient’s interaction force, leading to an inaccurate assessment of the patient’s true movement intent. To address this limitation, future studies may consider developing a more comprehensive control approach that incorporates the patient’s interaction force into the controller design.

Martinez et al. [37] proposed a velocity field based on the AAN to assist individuals in tracking their desired position and speed. This method defines an expected velocity at each point in Cartesian space, encodes a task trajectory in the end effector of an exoskeleton, and provides assistive force based on the patient’s current position and velocity. Hamed et al. [38] defined a path-tracking task by constructing an expected velocity field around the path in the velocity domain and designed two different controllers to adjust robot intervention based on user performance. Although these approaches allow users to adjust the task speed by adjusting the applied energy, variable task speeds can lead to a poor user experience and excessive assistive force, which may hinder patient motivation. 

In this paper, we propose and implement a fuzzy adaptive passive control strategy for an end-effector bilateral upper limb rehabilitation robot (EBULRR) based on task performance and impulse evaluation. The controller adjusts the assisted force according to the patient’s task performance and exertion to achieve effective training of the patient’s arm. Our controller offers minimal intervention during the rehabilitation process to maximize patient initiative while ensuring safety and compliance in robot-assisted rehabilitation. Compared to previous studies, the main contributions of this research can be summarized as follows: (1) This study proposes a fuzzy adaptive passive controller based on a potential field that is designed according to the desired motion trajectory, which can provide a normal force pointing towards the desired trajectory. (2) The controller uses the subject’s task performance and impulses as evaluation indices of the subject’s motor ability. (3) The controller adaptively adjusts the stiffness coefficient of the potential field using fuzzy logic, thereby adaptively changing the magnitude of the assisting force.

## 2. Materials and Methods

### 2.1. Mechanical Design of the EBULRR

The design of the EBULRR platform is shown in Figure 1 [39], which has three degrees of freedom and is capable of conducting rehabilitation movements in three-dimensional space. The EBULRR consists of two mechanical arms, one of which is named the Healthy Side Manipulator (HSM) and the other is named the Affected Side Manipulator (ASM). Although the mechanical mechanisms of both arms are fundamentally identical, the ASM has three joint actuators (Kollmorgen RGM14A/RGM17A) and is equipped with a 3D force sensor (Zlm1826) at its end-effector, while the HSM has no joint actuators or force sensor at its end-effector and can only be manipulated by the patient. During rehabilitation training, the patient can adjust their posture to the appropriate position, place their arms on the trays at the end-effectors of the mechanical arms, and secure them with nylon straps.

The design of the control system must rely on hardware components to be realized, mainly the upper computer, lower computer, motors, force sensors, data acquisition cards, etc. Figure 2 shows the hardware schematic of the control system in this paper, which mainly consists of a human–computer interaction module, a control module, and a data acquisition module. The main function of the control module is to communicate with the upper computer through the TCP/IP protocol to transmit interactive information and to communicate with the data acquisition card through the EtherCAT protocol to achieve real-time control of the robot. The data acquisition module mainly collects data from ASM end force sensors and motor encoders and drives through the data acquisition card, feeding this information to the lower computer. The human–robot interaction module is mainly used for the operation of the virtual environment, and studies have shown that combining virtual reality (VR) and rehabilitation robotics can enhance the training interest of patients and stimulate their active participation, which in turn is expected to improve the rehabilitation effect [40]. The study in this paper used Unity 2020.1.9f1c1 as the platform for designing the virtual environment, and the Unity virtual environment was projected onto the monitor during the experiment. The control period in this paper is 0.004 s.

### 2.2. Kinematic Modeling of the EBULRR

The kinematics of the robot is intended to describe the mapping relationship between the joint space of the robot and the cartesian space at the end of the robot. To facilitate the establishment of the kinematics of the robot, the origin of the base coordinate system is set at joint 1. In this paper, the standard D-H parameter method is used for kinematic analysis, and the coordinate system and D-H parameters of each linkage of the robot arm are shown in Figure 3 and Table 1, respectively.

### 2.3. Dynamics of the EBULRR

The dynamics model of the robot arm can be obtained by the Lagrange–Euler equation, and the dynamics equation of the robot can be described as:(1)$$M\left(q\right)\ddot{q}+C\left(q,\dot{q}\right)\dot{q}+G\left(q\right)+f\left(\dot{q}\right)=\tau +J^{T}\left(q\right)F_{ext}$$
where $q\in R^{n}$ is the joint vector and n is the number of joints, $M\left(q\right)\in R^{n\times n}$ denotes the inertia matrix, $C\left(q,\dot{q}\right)\in R^{n\times n}$ denotes the centripetal force and Coriolis matrix, $G\left(q\right)\in R^{n}$ is the gravity matrix, $f\left(\dot{q}\right)\in R^{n}$ denotes the friction matrix, $\tau \in R^{n}$ denotes the moment vector applied by the actuator, $F_{ext}\in R^{n}$ denotes the contact force generated by external application at the end of the robot arm, and $J\left(q\right)$ denotes the Jacobi matrix.

The joint space control law can be expressed as:(2)$$\tau_{a}=J^{T}\left(q\right)\left(-\nabla \Phi \left(s\right)-\Gamma \left(s,\dot{s}\right)\right)$$
where s denotes the position of the ASM end in Cartesian space, $\Phi \left(s\right)$ denotes the scalar constant potential function, $\nabla \Phi \left(s\right)$ denotes the gradient of the scalar constant potential function, $\Gamma \left(s,\dot{s}\right)$ denotes the dissipative field, and the design of the potential field and dissipative field will be introduced in the next subsection.

Thus, the robot dynamics equation can be rewritten as:(3)$$M\left(q\right)\ddot{q}+C\left(q,\dot{q}\right)\dot{q}+G\left(q\right)+f(\dot{q})=\tau +J^{T}\left(q\right)(-\nabla \Phi \left(s\right)-\Gamma \left(s,\dot{s}\right)+F_{ext})$$

### 2.4. Design of Potential Field and Dissipative Field

To achieve assisted rehabilitation training of the upper extremity, this paper constructs a potential field on the desired trajectory. During the training process, a normal force pointing to the desired trajectory is provided by the deviation between the patient’s hand and the desired trajectory, and the larger the deviation of the trajectory, the larger the assisted force that needs to be applied to the potential field. The potential energy of the desired trajectory is globally equal and minimal, i.e., the potential field gradient is zero at the desired trajectory. Therefore, the potential energy field can be designed as follows:

*N* points are uniformly sampled from the designed trajectory:(4)$$D_{s}={\left\{s_{r}^{i}\right\}}_{i=1}^{N}$$

$s_{r}^{i}\in R^{3}$ denotes the position information of the end of ASM at the *i*th point in the Cartesian space. $D_{s}$ is the discrete desired trajectory dataset of ASM.

The energy element at point s can be expressed as:(5)$$\phi^{i}\left(s\right)=\phi_{0}^{i}+\frac{1}{2}{\left(s-s_{r}^{i}\right)}^{T}K\left(s-s_{r}^{i}\right)\;\forall i\in 1\ldots N$$

The energy element $\phi^{i}\left(s\right)$ is determined jointly by the position of the sampling point $s_{r}^{i}$ and the position of point s. $\phi_{0}^{i}$ is a constant scalar and K is the potential field stiffness coefficient. For each energy element $\phi^{i}\left(s\right)$, the force at which point s is attracted to the sampled point $s_{r}^{i}$ is given by $-K\left(s-s_{r}^{i}\right)$. From Equation (5), it can be seen that the larger the value of K and the greater the distance between point s and sampling point $s_{r}^{i}$, the greater the attraction.

Calculate the weight of the potential energy at point s to the *N* sampling points using the Gaussian kernel function:(6)$$\omega^{i}\left(s\right)=e^{-\frac{1}{2{\left(\sigma^{i}\right)}^{2}}{\left(s-s_{r}^{i}\right)}^{T}\left(s-s_{r}^{i}\right)}\;\forall i\in 1\ldots N$$
where $\sigma^{i}$ is the smoothing parameter that controls the region of influence of each energy element. The total potential energy at s is given by:(7)$$\Phi \left(s\right)=\frac{\sum_{i=1}^{N}\omega^{i}\left(s\right)\phi^{i}\left(s\right)}{\sum_{j=1}^{N}\omega^{j}\left(s\right)}$$

Define the normalized weights ${\tilde{\omega }}^{i}(s)$:(8)$${\tilde{\omega }}^{i}\left(s\right)=\frac{\omega^{i}\left(s\right)}{\sum_{j=1}^{N}\omega^{j}\left(s\right)}\;\forall i\in 1\ldots N$$

The total potential energy of point s can be expressed as:(9)$$\Phi \left(s\right)=\sum_{i=1}^{N}{\tilde{\omega }}^{i}\left(s\right)\phi^{i}\left(s\right)$$

In the potential field, the potential energy is the same at each sampling point. The gradient of the potential field:(10)$$\nabla \Phi \left(s\right)=\sum_{i=1}^{N}\frac{1}{{{(\sigma }^{i})}^{2}}{\tilde{\omega }}^{i}\left(s\right)(\phi^{i}\left(s\right)-\Phi \left(s\right))\left(s-s_{r}^{i}\right)-{\tilde{\omega }}^{i}\left(s\right)K\left(s-s_{r}^{i}\right)$$

Using the circle as the desired trajectory, the gradient of the designed potential energy field on the sampled data set $D_{s}$ is expected to be zero. Therefore, the choice of the potential field parameter $\phi_{0}^{i}$ can be translated into solving the convex optimization problem with the following equation:(11)$$\begin{array}{l} \min J\left(\Lambda \right)=\frac{1}{N}\sum_{i=1}^{N}{\left\|\nabla \Phi \left(s_{r}^{i};\Lambda \right)\right\|}^{2}\\ subject\;to\\ 0\leq \phi_{0}^{i}\;\;\forall i\in 1\ldots N\\ \nabla \Phi \left(s_{r}^{i}\right)=0\;\forall i\in 1\ldots N\\ \Lambda =\left\{\phi_{0}^{1}\ldots \phi_{0}^{N}\right\} \end{array}$$

The designed potential field is shown in Figure 4. For more information about the potential field, please refer to the literature [41]. The design of the dissipative field is as follows:(12)$$\gamma^{i}\left(\dot{s}\right)=B^{i}\dot{s}$$
where $B^{i}\in R^{3\times 3}$ is a positive definite matrix and the total dissipative energy can be obtained from the nonlinear weighting of each dissipative unit $\gamma^{i}\left(\dot{s}\right)$:(13)$$\Gamma \left(s,\dot{s}\right)=\sum_{i=1}^{N}{\tilde{\omega }}^{i}\left(s\right)\gamma^{i}\left(\dot{s}\right)$$

### 2.5. Design of FAP

The proposed FAP controller in this paper utilizes fuzzy logic to adaptively modify the stiffness of the potential field, thereby changing the magnitude of the assisting force. Fuzzy logic employs the subject’s task performance and impulse as inputs, with the calculation formula as follows:(14)$${Error}_{i}=\sqrt{{\left(x_{i}-x_{0}\right)}^{2}+{\left(y_{i}-y_{0}\right)}^{2}}-R$$
(15)$$Score=100-\sqrt{\sum_{i=1}^{N}\frac{{{Error}_{i}}^{2}}{N}}$$
(16)$$I_{a}=\int_{0}^{t_{1}}Fdt$$
where ${Error}_{i}$ represents the error of deviation from the target trajectory at the end of the robot at the current ith point, $x_{i},y_{i}$ represents the coordinates of the current ith point, $x_{0},y_{0}$ represents the coordinates of the center of the circle, R represents the radius of the circle, Score represents the score of one revolution of motion, N represents the number of sampling points of one revolution of motion, $I_{a}$ represents the impulse of one revolution of motion, and $t_{1}$ represents the time of one revolution of motion.

The design of the stiffness-adaptive law K for the potential field adheres to the following principles: In the case of a patient with limited physical capacity, characterized by a low task score and a large impulse, an increase in the stiffness coefficient K of the potential field is warranted, as it indicates that the patient is exerting significant effort but is still unable to complete the task satisfactorily. Conversely, if the patient’s physical ability improves and they can achieve high scores in the task, or if the patient becomes demotivated during training, resulting in a small impulse, a reduction in the stiffness coefficient K of the potential field is recommended to encourage the patient’s proactivity during the training process.

Based on the above principles, fuzzy logic rules were designed and used as evaluation algorithms with evaluation indicators Score and $I_{a}$ as inputs to quantitively assess the subject’s movement ability, thus adaptively modifying the stiffness coefficient K of the potential field and thus changing the magnitude of the assistance force to stimulate the subject’s initiative. The fuzzy logic system mainly consists of four stages: defining input and output, fuzzification, fuzzy inference based on rule base, and defuzzification. Its structure is shown in Figure 5. 

(1) Defining the inputs and outputs: The average score (Score) of the subjects in the ith training task and the impulse ($I_{a}$) of the subjects during one revolution of motion are chosen as the two inputs of the fuzzy logic system. To ensure that these inputs are in a suitable range for the fuzzy logic system, scaling factors $k_{s}$ and $k_{i}$ are used to scale the input variables to the universe of discourse as [60, 95] and [0.5, 1.5], respectively. The output of the fuzzy logic system is the stiffness coefficient (K) of the potential field, which is also scaled to the universe of discourse as [0, 12]. 

(2) Perform fuzzification: Perform fuzzification to obtain the input and output membership functions. The inputs and outputs are fuzzified by the membership functions shown in Figure 6. Clear input and output values are converted into linguistic variables and membership values. The fuzzy set of each input or output is determined by the membership function. The membership function of a fuzzy set is defined by five triangular functions. The fuzzy set of each input is named very small (VS), small (S), medium (M), big (B), and very big (VB), and the fuzzy set of the output linguistic variables is named zero (ZO), positive small (PS), positive medium (PM), and positive big (PB).

(3) Inference process: This process generates the membership degrees of the output linguistic variables based on the fuzzified inputs. The key to the inference process is the establishment of fuzzy rules. In the fuzzy inference stage, the “If-Then” type of language is used based on fuzzy rules. Table 2 provides an exhaustive list of the fuzzy outputs generated by all 25 input combinations. The fuzzy rules are based on the above principles and are further improved by experiments. The inference process is based on the Mamdani inference method. The input and output surface diagrams of the fuzzy logic are shown in Figure 7.

(4) Defuzzification is the conversion of the inferred fuzzy values into explicit control signals as input values to the system. 

The FAP controller based on the above design can adaptively adjust the potential field stiffness coefficient K, and thus the magnitude of the assisted force, to suit subjects with different motor abilities. The effect of the potential field stiffness coefficient K on the potential field is shown in Figure 8.

The simplified control block diagram of FAP proposed in this paper is shown in Figure 9.

### 2.6. Proof of Stability

The FAP control strategy proposed in this paper is stable because it is modeled by potential functions and dissipative fields [41]. The stability has been demonstrated through the subsequent theorem and derivation process.

The robot dynamics equation can be defined as follows:(17)$$M\left(x\right)\ddot{x}+C\left(x,\dot{x}\right)\ddot{x}+G\left(x\right)+f(\dot{x})=\tau +\tau_{ext}$$
where $x\in R^{n}$ is the state vector, either joint space or Cartesian space, n is the number of joints, $M\left(x\right)\in R^{n\times n}$ denotes the inertia matrix, $C\left(x,\dot{x}\right)\in R^{n\times n}$ denotes the centripetal and Coriolis matrices, $G(x)\in R^{n}$ is the gravity matrix, $f\left(\dot{x}\right)\in R^{n}$ is the friction matrix, $\tau \in R^{n}$ denotes the moment vector applied by the actuator, and $\tau_{ext}\in R^{n}$ denotes the contact force applied externally at the end of the robot arm.

The torque τ applied by the actuator consists of the control term $\tau_{a}$, the robot gravity compensation term, and the frictional force compensation term:(18)$$\tau =\tau_{a}+G\left(x\right)+f(\dot{x})$$

The control term $\tau_{a}$ consists of the potential field gradient $\nabla \Phi \left(x\right)$ and the dissipative field $\Gamma \left(x,\dot{x}\right)$:(19)$$\tau_{a}=-\nabla \Phi \left(x\right)-\Gamma \left(x,\dot{x}\right)$$

We will prove the stability of the controller using the passivity formalism [38], giving the following theorem:

**Theorem 1.** *A system with input effort* u *and output flow* y *is passive if it satisfies:*(20)$$\dot{V}=y^{T}u-g$$

A system that satisfies Equation (20) and has a lower bound on V with g≥0 is passive/stable.

**Proof.** Set the effort to $\tau_{ext}$ and the flow to x. To ensure the passive/stability of the controller, we define the following candidate Lyapunov functions:
(21)$$V\left(x,\dot{x}\right)=\Phi \left(x\right)+\frac{1}{2}{\dot{x}}^{T}M\left(x\right)\dot{x}$$Differentiating Equation (21) yields:(22)$$\dot{V}\left(x,\dot{x}\right)={\dot{x}}^{T}\nabla \Phi \left(x\right)+{\dot{x}}^{T}M\left(x\right)\ddot{x}+\frac{1}{2}{\dot{x}}^{T}\dot{M}\left(x,\dot{x}\right)\dot{x}$$$M\left(x\right)\ddot{x}$ can be obtained using Equation (17):(23)$$M\left(x\right)\ddot{x}=\tau +\tau_{ext}-C\left(x,\dot{x}\right)\ddot{x}-G\left(x\right)-f(\dot{x})$$Bringing Equations (18) and (19) into Equation (23) yields:(24)$$M\left(x\right)\ddot{x}=-\nabla \Phi \left(x\right)-\Gamma \left(x,\dot{x}\right)+\tau_{ext}-C\left(x,\dot{x}\right)\ddot{x}$$Then, bringing Equation (24) into Equation (22) yields:(25)$$\dot{V}\left(x,\dot{x}\right)=-{\dot{x}}^{T}\Gamma \left(x,\dot{x}\right)+{\dot{x}}^{T}\tau_{ext}+{\dot{x}}^{T}[\frac{1}{2}\dot{M}\left(x,\dot{x}\right)-C\left(x,\dot{x}\right)]\dot{x}$$Since $\dot{M}\left(x,\dot{x}\right)-2C\left(x,\dot{x}\right)$ is a skew-symmetric matrix, Equation (25) can be simplified to:(26)$$\dot{V}\left(x,\dot{x}\right)={\dot{x}}^{T}\tau_{ext}-{\dot{x}}^{T}\Gamma \left(x,\dot{x}\right)$$Finally, bringing Equations (12) and (13) into Equation (26) yields:(27)$$\begin{array}{ll} \dot{V}\left(x,\dot{x}\right) & ={\dot{x}}^{T}\tau_{ext}-{\dot{x}}^{T}(\sum_{i=1}^{N}{\tilde{\omega }}^{i}\left(x\right)B^{i})\dot{x}\\ & =y^{T}u-g \end{array}$$Since ${\tilde{\omega }}^{i}\left(x\right)>0$ and $B^{i}$ is a positive definite matrix, g≥0. According to Theorem 1, it is known that this control system is passive/stable. □

## 3. Results

### 3.1. Security Verification

In the realm of rehabilitation robotics, ensuring the safety of the robot is of utmost importance since a significant proportion of robot users are individuals suffering from physical disabilities and functional impairments. It is essential to prevent any potential harm or secondary injuries that may occur during the use of the robot. Conventional impedance controllers allow the robot arm to track changing target positions within a certain range of tracking errors. However, external factors, such as sudden spasticity in the patient, can cause the robotic arm to abruptly halt during rehabilitation training. This occurrence may result in the impedance controller generating excessive force that could lead to secondary injuries.

Figure 10 is a comparison between the safety simulations of the FAP controller proposed in this paper and a traditional impedance controller. At 5 s, the simulation subject experienced a spasm, causing the robotic arm to stop. Under different controllers, the robotic arm end will produce different forces. The blue dashed line in the figure represents the expected trajectory of the robotic arm in the X direction; the blue solid line represents the actual trajectory of the robotic arm in the X direction; the red dashed line represents the force generated by the traditional impedance controller; and the red solid line represents the force generated by the FAP controller. It can be observed that both controllers provide only small assistive forces in the first five seconds. However, after the robotic arm suddenly stops in the last five seconds, the impedance controller will generate a large force, which could be dangerous for the user. However, because the force provided by the FAP controller is only dependent on the current position, the force generated by FAP throughout the process is small and will not harm the user in the event of an unexpected situation. Therefore, it can be concluded that the FAP controller proposed in this paper is safe for use in rehabilitation training.

### 3.2. The Effect of the FAP Controller on Subjects’ Task Trajectory and Initiative

For patients in the middle to late stages of rehabilitation who have regained some upper extremity motor abilities, it is critical that the robot only assists the patient if he or she is unable to complete the task on a predetermined trajectory, rather than completely preventing the patient from completing the task. Depending on the patient’s functional ability, it is better to provide only the necessary assistance or correction to allow the patient to be more active in completing the rehabilitation training tasks. Since the FAP controller does not provide assistance forces in the tangential direction of the target trajectory, this means that the patient can be motivated to exert more effort during rehabilitation training.

To investigate the effect of the FAP controller on the subject’s task trajectory and initiative, we will design two modes, passive mode and active mode, and the FAP controller for comparison experiments. The control strategy of the passive mode consists of robot inverse dynamics feedforward plus a PD controller to track the target trajectory, in which the robot drives the subject completely to complete the task. The active mode control strategy provides only gravity compensation and friction compensation for the robot, and the subject completes the task by himself in this mode.

During the experiment, the subject was holding an ASM end-effector and needed to complete a two-dimensional planar circle drawing task based on the visual feedback provided by the display, as shown in Figure 11. The coordinates of the initial point of the task were (0.68, 0.1, 0) m; the coordinates of the circle’s center were (0.68, 0, 0) m; the radius was 0.1 m; and the circle was located in the XY plane. The subjects were required to complete the circle drawing task 10 times in each of the 3 modes: FAP, passive, and active. The FAP controller parameters were set to N=124,σ=0.01. The PD controller parameters for passive mode were set to $k_{p}=\left[40\;50\;30\right],k_{d}=[1\;2\;1]$.

The experiments were conducted by eight healthy volunteers (five males and three females, aged 24–34 years). The experiment was conducted in the Rehabilitation Robotics Laboratory of the Ningbo Institute of Materials Technology and Engineering, Chinese Academy of Sciences, and approved by the Ethics Committee of Cixi Institute of Biomedical Engineering, Ningbo Institute of Materials Technology and Engineering, Chinese Academy of Sciences. Before starting the trial, each subject was fully informed about the trial procedure and signed an informed consent form.

Figure 12 shows some of the mission trajectory plots of S1 in the three modes, and it can be seen that the error in tracking the target trajectory is smaller in the passive mode and the mission trajectory error is the largest in the active mode. Figure 13 shows the task trajectory diagram of S1 in the FAP mode. The red line indicates the desired trajectory of the rehabilitation task, and the blue line indicates the subject’s three task trajectories in the FAP mode. It can be seen that in the initial task, the subjects’ actual trajectories deviated significantly from the desired trajectories. As the number of tasks increases, the deviation of the subjects’ trajectories gradually decreases. This was caused by the low initial value of the potential field stiffness factor K, the small assistance force provided by the FAP controller, and the subject’s lack of proficiency. However, as the potential field stiffness coefficient K became larger, the assisted force provided by the FAP controller also became larger, and the deviation of the subjects’ trajectories gradually decreased. 

Figure 14 shows the evolution of $Score,I_{a}$, and K during S1’s task in FAP mode. In this figure, the subject’s ability was improved, and the task score gradually increased due to the increase in the number of training sessions; the impulse was also maintained at a certain level, which indicated that the subject had been putting in effort during the task. Therefore, the stiffness coefficient K of the potential energy field gradually decreases, thus minimizing the assistance force and fully motivating the subject’s initiative.

To investigate the effect of the FAP controller on the subjects’ initiative, we analyzed the subjects’ interactive forces during the circle drawing process, and Figure 15a shows the comparison of the force sensor values at the end of the robot for eight subjects (S1–S8) in different training modes. From the figure, it can be seen that the values of force sensors in FAP mode are much larger than those in passive mode and are similar to those in active mode. Figure 15b shows the group average values of the force sensor for the eight subjects in active, passive, and FAP modes, which were (5.5 ± 2.23) N, (17.73 ± 4.60) N, and (18.68 ± 5.77) N. This indicates that when performing a circular drawing task in FAP mode, the subjects did not exhibit any slackening due to the assistance provided by the robot, as was observed in the passive mode. Instead, they exerted comparable effort to that in the active mode and demonstrated greater initiative.

### 3.3. Experiment on the Effectiveness of FAP Controller for Motor Learning

To demonstrate that the proposed FAP controller is effective for motor learning, a crossover experiment is designed to compare which mode of KA and KF is better for motor learning, where the potential field stiffness coefficient K of the KA mode is adaptively updated by fuzzy rules in the range of [0, 12], i.e., the higher the subject’s score, the smaller the impulse, the smaller the value of K; the lower the subject’s score, the larger the impulse, the larger the value of K. Additionally, the potential field stiffness coefficient K of the KF mode is fixed and takes the value of 12. Before the experiment starts, the subjects are instructed on how to complete the circle drawing task. The experiment was conducted by eight healthy volunteers (six males and two females, aged 24–34 years), and the subjects were randomly and equally divided into two groups (A and B) before the experiment.

The subjects were required to conduct the experiment for two days, and the first day of the experiment was divided into two stages: (1) the training stage, in which the potential field provided an assistance force to help the subjects complete the training, where group A performed a 20-turn circle drawing task in KA mode and group B performed a 20-turn circle drawing task in KF mode, and (2) the evaluation stage, in which the subjects rested for one minute after completing the training stage, during which the robot did not provide assistance force, and both groups A and B were required to perform 10 revolutions of the circle drawing task.

After a three-day washout period, subjects underwent a second experiment: (1) At the training stage, in contrast to the first day, group B performed a 20-turn circle drawing task in KF mode, and group A performed a 20-turn circle drawing task in KF mode. (2) At the evaluation stage, the same as the first day, both groups A and B were required to perform 10 revolutions of the circle drawing task. This was to eliminate the effect of the order on the experiment. We abbreviated the training stage with the circle drawing task in KA mode as KATS, the training stage with the circle drawing task in KF mode as KFTS, the evaluation stage with the circle drawing task in KA mode as KAES, and the evaluation stage with the circle drawing task in KF mode as KFES.

Figure 16a shows the comparison of the actual trajectory of S1 in the training stage, and it can be seen that the trajectory error of S1 in KFTS is lower than that in KATS. Figure 16b shows the comparison of the actual trajectory of S1 in the evaluation stage, and in contrast, the trajectory error of S1 in KFES is higher than that in KAES. Figure 17 shows the evolution of the task score of S1 as the number of tasks increases, and it can be seen that the task score in KFTS is higher than that in KAES because the potential field stiffness coefficient is larger and provides a larger assistance force, so the task score is higher than that of KATS. However, after removing the assistance force in the evaluation stage, S1’s score is higher in KAES than in KFES.

Figure 18 shows a graph of the mean scores of the eight subjects at different stages, from which it can be seen that the mean score of KFTS is slightly higher than that of KATS for the eight subjects during the training stage, but the score of KAES is higher than that of KFES during the evaluation stage.

A paired-sample *t*-test was conducted on the results of two experimental groups, KA and KF. The group mean scores of eight subjects are presented in Table 3. During the training stage, it was found that the group mean scores of subjects were higher in the KF mode, and there was a significant difference between the two modes (p<0.001). However, in the subsequent evaluation stage, where the robot did not provide any assistance and subjects had to complete the task entirely on their own, it was observed that subjects achieved higher group mean scores after training in the KA mode, and there was a significant difference between the two modes (p<0.001). This indicates that after KF mode training, subjects may have developed a slack mentality and felt that they could do the task well without effort. In contrast, after the KA mode training, the subjects could still complete the task well without the assistance force, and they had higher initiative. 

Figure 19a shows the comparison graph of K values of S1 in KA mode and KF mode, and it can be seen that the K values of S1 in KA mode are much smaller than those in KF mode. Figure 19b shows the comparison plots of K values for eight subjects in KA mode and KF mode, and the results show that all eight subjects have much smaller K values in KA mode than in KF mode. This may explain the subjects’ lower task scores in KATS than in KFTS because the subjects had smaller K values in KATS and the robot provided less assistance force. In contrast, in the evaluation phase, with the withdrawal of the assistance force, subjects had higher task scores in KAES than in KFES. This indicates that the subjects were able to perform the task well without the assistance of the robot after being trained in KA mode. Therefore, it can be proved that the FAP control strategy proposed in this paper has a good motor learning effect.

## 4. Discussion

With the development of science and technology, the application of robotics has widely increased in various industries, and there are more and more examples of robots moving out of the factory environment and joining the real-life human environment, so human–robot interaction technology [42] has been heavily researched. For example, the creation of rehabilitation robots has enabled a large number of stroke patients to be trained effectively. However, while rehabilitation robots bring help to people, they also bring hidden dangers and may cause secondary injuries to patients. Therefore, a core issue of human–robot interaction is to ensure the safety of the user [43]. As shown in Figure 10, if the robotic arm stops suddenly due to external reasons during the rehabilitation training, e.g., if the patient suddenly has a spasm phenomenon, then the traditional impedance control will generate a large moment, which may cause secondary injuries to the patient, while the FAP control strategy in this paper will stop assistive force to ensure the safety of the patient.

In addition, the majority of patients tend to slack off and take little initiative with the help of rehabilitation robots, which leads to poor rehabilitation results. Therefore, there is a need to develop an adaptive AAN controller that assists only when the subject needs help and stimulates the subject’s subjectivity. Most current on-demand assistive controllers can only vary the amount of assistive force based on task performance, which can ignore the patient’s motor intentions. Additionally, some controllers can provide too much assistance, thus reducing the patient’s initiative. The FAP controller proposed in this paper can adaptively adjust the size of the assistance force through fuzzy rules based on the patient’s task performance and impulse, providing minimal intervention during the rehabilitation process and fully stimulating the patient’s initiative. As shown in Figure 15, the FAP controller proposed in this paper still requires a lot of effort from the subject despite the provision of assisted force, thus preventing the subject from slacking off and taking high initiative.

Motor learning is crucial for humans to interact with their surroundings and independently perform activities of daily living, and it has long been recognized that the fields of motor learning and neurological rehabilitation are interconnected and that a full understanding of human motor learning can help patients with neurological injuries achieve better rehabilitation [44]. In this paper, we experimentally demonstrate that the FPA controller is helpful for subjects’ motor learning, as shown in Figure 17 and Figure 18, where subjects trained with the FPA controller can perform the circle drawing task well without the help of assistive forces.

In addition, after discussion with clinical therapists, it is found that Brunnstrom stages are very commonly adopted clinically to classify stroke patients’ rehabilitation progress due to its simplicity and validity. The method divides the motor recovery process into six stages, from the period of complete flaccidity, which begins immediately after stroke, to the disappearance of spasticity when the patient is able to perform near-normal to normal movement [45]. Each stage requires a different approach to rehabilitation training. For example, in the first two stages, a passive mode can be designed in which the patient is completely guided by the robot to complete the rehabilitation training; in the third and fourth stages, a robot-assisted mode can be designed to provide assistance force to help the patient complete the rehabilitation training; and in the fifth and sixth stages, an active mode or even a resistance mode can be designed to allow the patient to complete the rehabilitation training. The FAP controller in this paper is suitable for patients in stages three and four because patients in this stage have partial motor ability, and we can provide assisted forces to help them when they deviate from their trajectory but also to allow them to complete the task autonomously near the task trajectory. Therefore, in the future, the algorithm for the evaluation needs to be different for patients at different stages, i.e., the adaptive law of the potential field stiffness coefficient needs to be improved. Moreover, more adaptive control algorithms can be designed to meet the training needs of patients at each stage and improve their rehabilitation effectiveness.

## 5. Conclusions

In response to the issue that most rehabilitation robot control strategies can lead to low patient initiative and thus affect rehabilitation effectiveness, this paper proposes a fuzzy adaptive passive control strategy based on task performance and impulse. To ensure the safety of the subjects, a passive controller based on the potential field is designed to guide and assist patients in their movements, and the stability of the controller is demonstrated in a passive formalism. Then, using the subject’s task performance and impulse as evaluation indicators, fuzzy logic rules were designed and used as an evaluation algorithm to quantitively assess the subject’s motor ability and to adaptively modify the stiffness coefficient of the potential field and thus change the magnitude of the assistance force to stimulate the subject’s initiative. Through experiments, this control strategy has been shown to not only improve the subject’s initiative during the rehabilitation process and ensure their safety during training but also enhance the subject’s motor learning ability. However, this paper has certain limitations in that only healthy subjects were recruited for experimental testing to validate the effectiveness of the proposed controller. In the next step, this controller is planned to be used for stroke patients to investigate its impact on patient rehabilitation effectiveness.

## Figures and Tables

**Figure 1 sensors-23-04042-f001:**
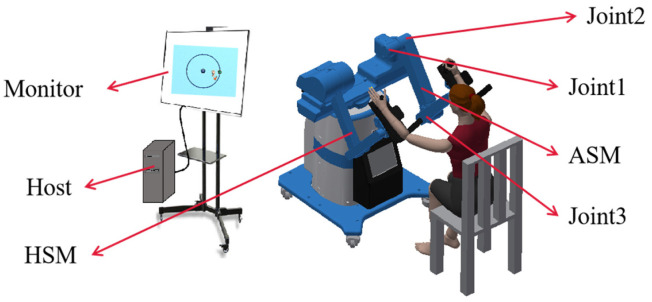
The design of the EBULRR platform.

**Figure 2 sensors-23-04042-f002:**
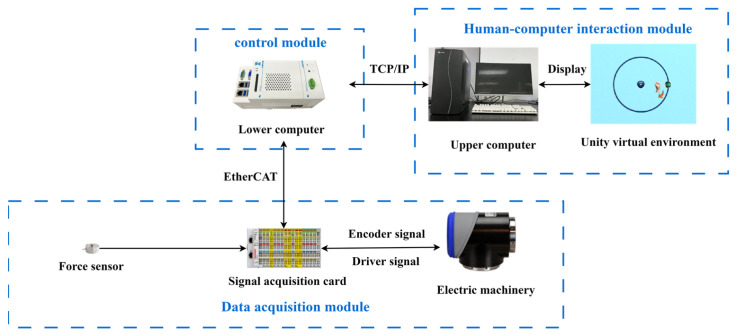
Diagram of the control system hardware.

**Figure 3 sensors-23-04042-f003:**
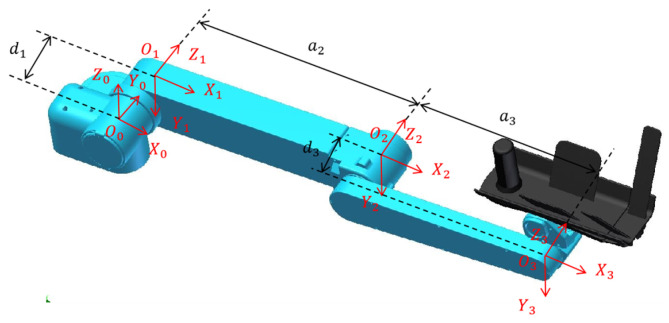
Diagram of the coordinate system of each linkage of the robot arm.

**Figure 4 sensors-23-04042-f004:**
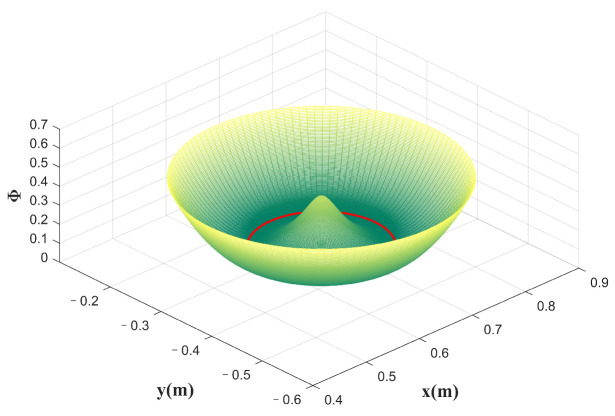
The potential energy field is designed by using a circle as the desired trajectory; the red line is the desired trajectory. The higher the potential energy at the current point, the greater the normal force provided.

**Figure 5 sensors-23-04042-f005:**
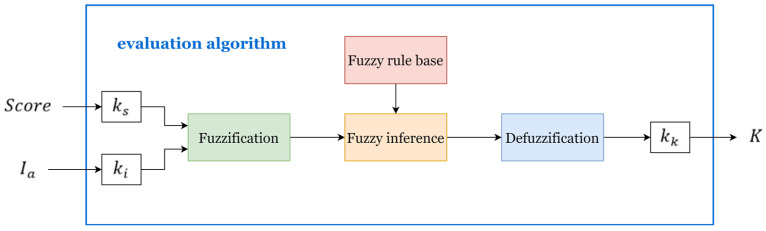
Fuzzy inference system structure diagram.

**Figure 6 sensors-23-04042-f006:**
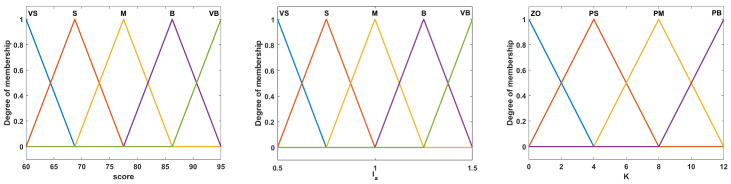
Membership functions for the input and output.

**Figure 7 sensors-23-04042-f007:**
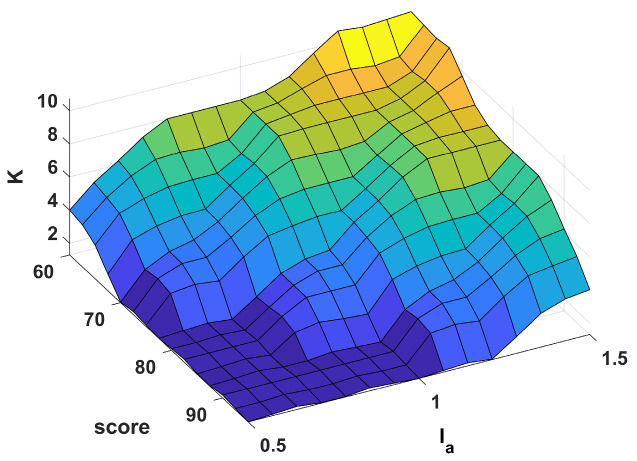
Input and output surface diagram of fuzzy logic.

**Figure 8 sensors-23-04042-f008:**
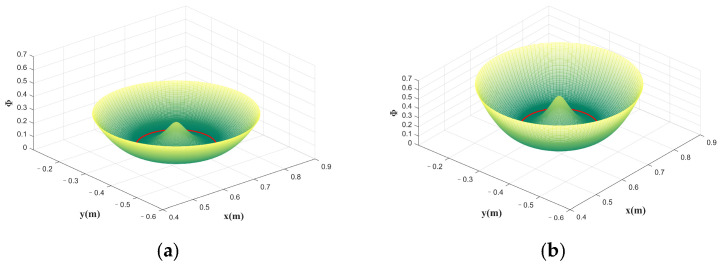
The effect of the potential field stiffness coefficient  K on the potential field. (**a**) K=6 (**b**) K=12. Higher potential energy provides greater assistance force.

**Figure 9 sensors-23-04042-f009:**
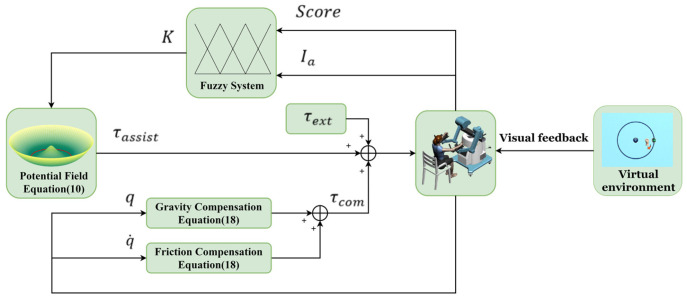
The simplified control block diagram of FAP.

**Figure 10 sensors-23-04042-f010:**
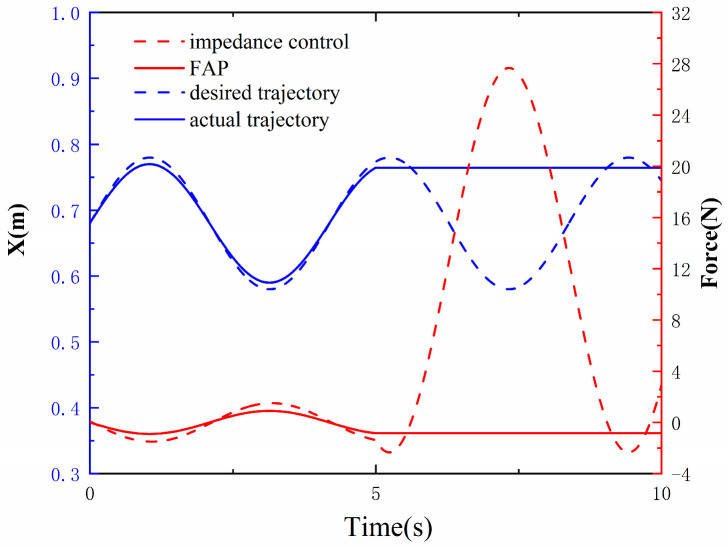
Comparison chart of security simulation tests between the proposed FAP controller and the conventional impedance controller.

**Figure 11 sensors-23-04042-f011:**
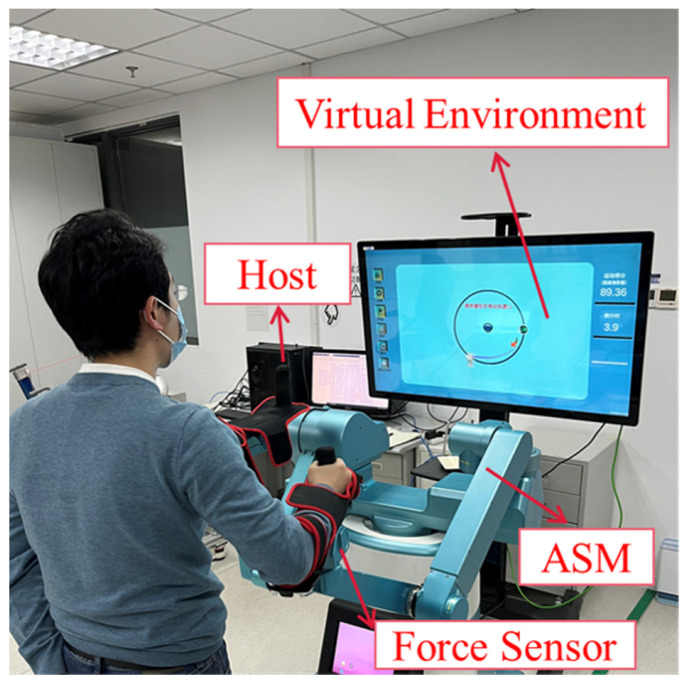
The subject was participating in an experiment with a circle drawing task.

**Figure 12 sensors-23-04042-f012:**
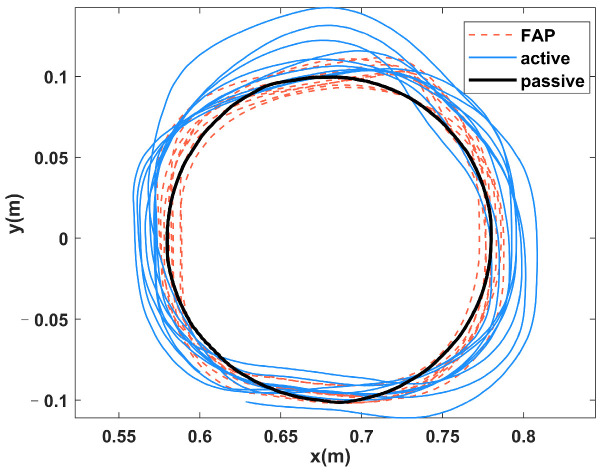
Partial task trajectory diagram of S1 in three modes.

**Figure 13 sensors-23-04042-f013:**
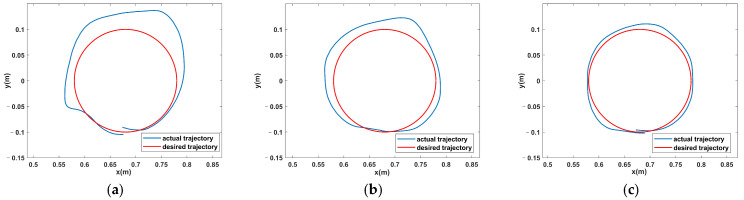
S1’s task trajectory diagram in FAP mode: (**a**) task1, (**b**) task7, and (**c**) task19.

**Figure 14 sensors-23-04042-f014:**
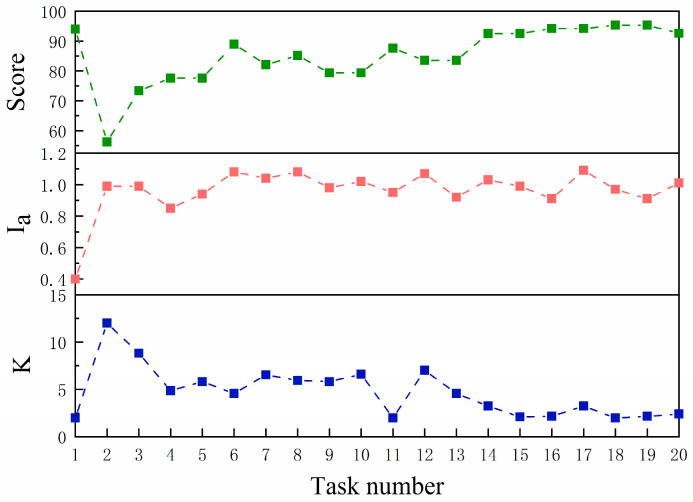
Evolution of $Score,I_{a}$, and K during S1’s task in FAP mode.

**Figure 15 sensors-23-04042-f015:**
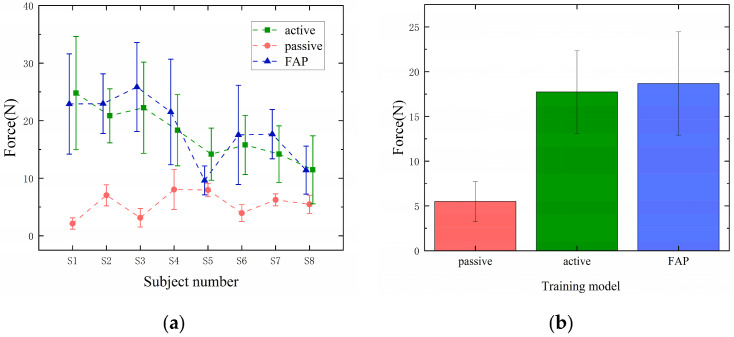
(**a**) Comparison of robot end force sensor values for eight subjects in different training modes. (**b**) Group average of force sensor values for eight subjects in different modes.

**Figure 16 sensors-23-04042-f016:**
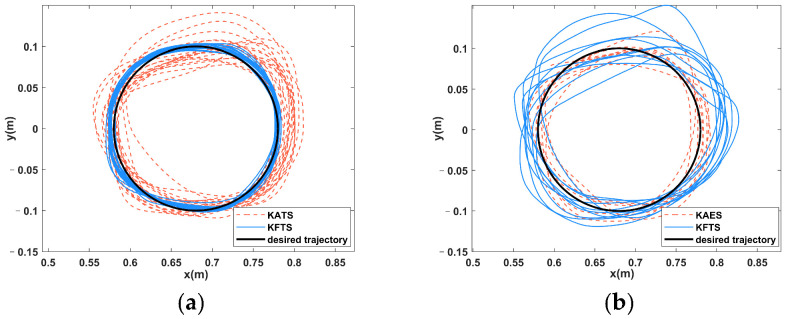
(**a**) Comparison chart of actual trajectory of S1 in training stage. (**b**) Comparison chart of actual trajectory of S1 in the evaluation stage.

**Figure 17 sensors-23-04042-f017:**
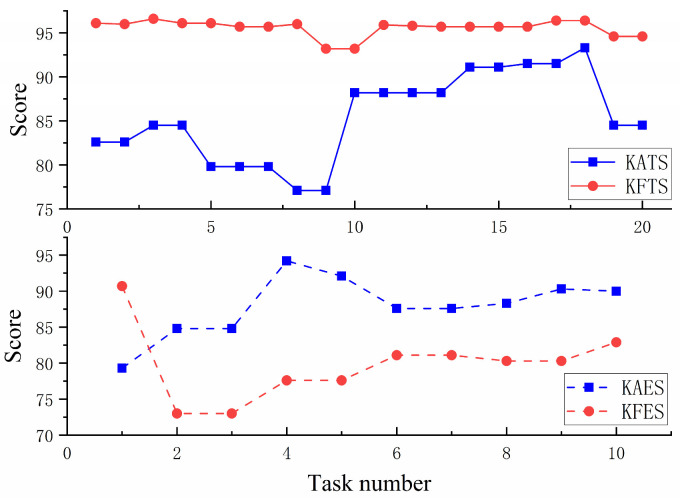
The evolution of the Score of S1 in different stages. The top half of the image shows the comparison of Score during the training stage and the bottom half of the image shows the comparison of scores during the evaluation stage.

**Figure 18 sensors-23-04042-f018:**
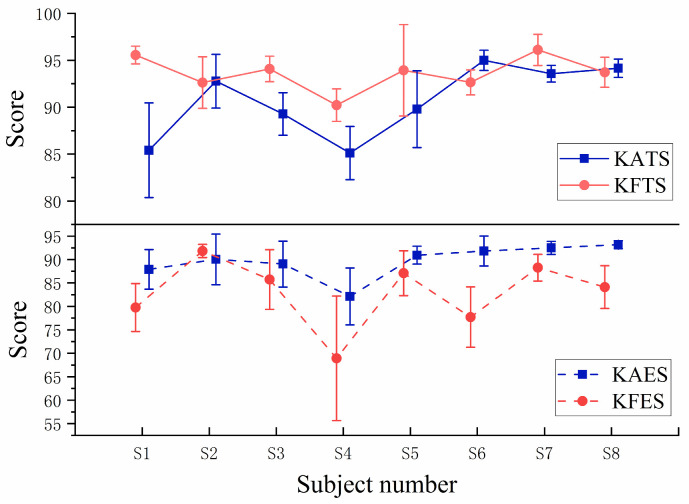
Graph of the mean scores of the eight subjects at different stages. The top half of the image shows the comparison of mean scores during the training stage and the bottom half of the image shows the comparison of mean scores during the evaluation stage.

**Figure 19 sensors-23-04042-f019:**
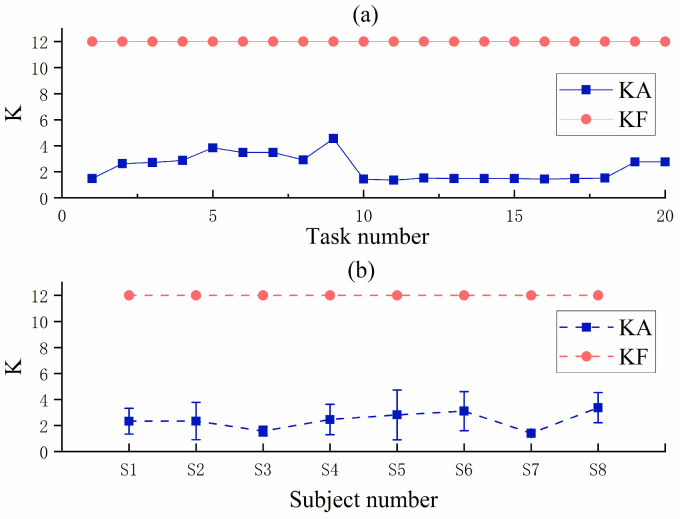
(**a**) Comparison of K values of S1 in different modes. (**b**) Comparison of K values of eight subjects in different modes.

**Table 1 sensors-23-04042-t001:** D-H parameters table of the EBULRR.

Link	$\theta_{i}/({}^{\circ})$	$\alpha_{i}/({}^{\circ})$	$a_{i}/(\mathrm{mm})$	$d_{i}/(\mathrm{mm})$
1	$\theta_{1}$	−90	0	68
2	$\theta_{2}$	0	500	0
3	$\theta_{3}$	0	420	−68

**Table 2 sensors-23-04042-t002:** Fuzzy rule table.

$Score/I_{a}$	VS	S	M	B	VB
**VS**	PS	PM	PM	PB	PB
**S**	ZO	PS	PM	PM	PB
**M**	ZO	ZO	PS	PM	PM
**B**	ZO	ZO	ZO	PS	PM
**VB**	ZO	ZO	ZO	ZO	PS

**Table 3 sensors-23-04042-t003:** This table displays the group mean scores of a sample consisting of eight subjects.

Stage	KA	KF	p
training stage	90.63±4.60	93.61±2.88	0.001
evaluation stage	89.69±5.02	82.93±9.22	0.001

## Data Availability

The original data contributions presented in the study are included in the article; further inquiries can be directed to the corresponding authors.

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
