# Peer review of "Fuzzy Adaptive Passive Control Strategy Design for Upper-Limb End-Effector Rehabilitation Robot"

_sensors, 2023, doi:10.3390/s23084042_

Round 1
Reviewer 1 Report
This study devoloped the FAP controller for the stroke patients' rehabilitation. Some comments are followed:
1. The comparison between KA and KF mode should be disclosed more detailes.
2. The key evaluation algorithm also need described clearly.
3. I am interested in the conclusions with the features of this study, such as the evaluation algorithm.
4. The minor suggestion is to discuss the evalution with a clinic person, like the therapists. Because the stroke patients appeared different situation at different stage of their recovery process. Please check the Brunnstrom Stages of Stroke Recovery.
Reviewer 2 Report
The article discusses the limitations of current rehabilitation robotic controllers and their inability to accurately assess a patient's motor intention and stimulate initiative, which negatively affects rehabilitation outcomes. To address this issue, the article proposes a fuzzy adaptive passive (FAP) control strategy that assesses a patient's task performance and impulse and adapts the stiffness coefficient and assistive force of the potential field accordingly. The topic is not original but some developments can be seen. Also others can use that as a paper for developing tracking control system. but they did not explain the challenged in the design and some important parameters such as workspace in the design. They need to improve the quality of the introduction and citations. They missed a lot:
Safety supervisory strategy for an upper-limb rehabilitation robot based on impedance control
Adaptive Fuzzy Sliding Mode Controller Design for a New Hand Rehabilitation Robot
Adaptive impedance control for upper-limb rehabilitation robot using evolutionary dynamic recurrent fuzzy neural network
Fuzzy sliding mode control of a wearable rehabilitation robot for wrist and finger
A new underactuated mechanism of hand tendon injury rehabilitation
